# Analysis of national and subnational prevalence of adolescent pregnancy and changes in the associated sexual behaviours and sociodemographic determinants across three decades in Ghana, 1988–2019

Shamsudeen Mohammed 

Department of Non-communicable Disease Epidemiology, London School of Hygiene & Tropical Medicine, London, UK

**Correspondence to**
Mr Shamsudeen Mohammed;
Shamsudeen.Mohammed1@lshtm.ac.uk

## ABSTRACT

**Objective** Understanding the determinants of adolescent pregnancy and how they have changed over time is essential for measuring progress and developing strategies to improve adolescent reproductive health. This study examined changes over time in the prevalence and determinants of adolescent pregnancy in Ghana.

**Methods** A total of 11 nationally representative surveys from the Ghana Demographic and Health Survey (1988, 1993, 1998, 2003, 2008, 2014), Multiple Indicator Cluster Survey (2006, 2011, 2017–2018) and Malaria Indicator Survey (2016 and 2019) provided data on 14 556 adolescent girls aged 15–19 for this analysis. A random-effect meta-analysis, time trends and multivariable logistic regression models were used to track the prevalence and determinants of adolescent pregnancy.

**Results** The pooled prevalence of adolescent pregnancy in Ghana was 15.4% (95% CI=13.49% to 17.30%). Rural areas (19.5%) had a higher prevalence of adolescent pregnancy than urban areas (10.6%). In the overall sample, middle adolescents (15–17 years) (aOR=0.30, 95% CI=0.23 to 0.39), adolescents in urban areas (aOR=0.56, 95% CI=0.43 to 0.74), large households (aOR=0.62, 95% CI=0.49 to 0.78), not working (aOR=0.62, 95% CI=0.43 to 0.90) and those unaware of contraceptive methods (aOR=0.49, 95% CI=0.27 to 0.90) were less likely to become pregnant. Adolescents from middle-income (aOR=0.91, 95% CI=0.67 to 1.24) or high-income (aOR=0.59, 95%CI=0.36 to 0.94) households, those who were semiliterate (aOR=0.56, 95%CI=0.39 to 0.82) or literate (aOR=0.28, 95%CI=0.21 to 0.37) and those with fewer previous sex partners were less likely to become pregnant. Not all determinants in the overall sample were consistently associated with adolescent pregnancy in the last three decades. Between 1988 and 1998, determinants of adolescent pregnancy were age, literacy, employment, household size and whether the mother was alive. Between 2003 and 2008, age, literacy, household size, income, age of last sexual partner, number of previous partners and contraception knowledge determined adolescent pregnancy. From 2011 to 2019, age, residence, literacy and menstrual cycle knowledge were determinants of adolescent pregnancy.

---

### STRENGTHS AND LIMITATIONS OF THIS STUDY

⇒ The study included 11 nationally representative surveys across 3 decades.

⇒ Because the sample size was large, the study had adequate power to detect and estimate more reliable results.

⇒ Adolescents from varied sociodemographic backgrounds were included in the study sample.

⇒ The secondary data sources lacked information on cultural and traditional practices that may influence adolescent pregnancy.

---

**Conclusion** Interventions and policies to prevent adolescent pregnancy should prioritise adolescents from disadvantaged backgrounds.

## BACKGROUND

Globally, one in five adolescent girls gives birth before age 18.[1] In low- and middle-income countries, about 21 million teenagers become pregnant each year,[2] and if current trends continue, that number is projected to triple by 2030.[1] The prevalence is highest in South Asia and sub-Saharan Africa.[1] In 2018, a systematic review and meta-analysis of 52 studies revealed that the prevalence of adolescent pregnancy in sub-Saharan Africa was 19.3%, with substantial regional variation ranging from 16% in Central Africa to 22% in East Africa.[3] The consequences of adolescent pregnancy can be devastating. For instance, teenagers who become pregnant face many negative social consequences, including isolation, stigma, rejection from peers and family, forced marriage and even the risk of dropping out of school.[4] Studies have shown that adolescents who become pregnant face an increased risk of unsafe abortion, eclampsia,

**BMJ**

obstruction during childbirth, obstetric fistula, preterm birth and maternal death or disability.[2 5–10]

Many epidemiological studies have shown a considerable disparity in the drivers of adolescent pregnancy, where disadvantaged adolescents are more at risk of pregnancy than those better off. For example, child marriage, non-school attendance, poverty, living in rural residence, household living arrangement, no maternal education, no paternal education, history of maternal teenage pregnancy, cost of contraceptives, misconceptions about contraceptives and lack of parental counselling and guidance on sexual and reproductive health have been identified as the main sociodemographic and economic factors that drive adolescent pregnancy in Africa.[3 11 12] Furthermore, sexual risk factors, including unwanted sexual advances from adult males, coercive sexual relations, unequal gender power relations, lack of comprehensive sexuality education and early sexual debut, also increase adolescents likelihood of becoming pregnant.[3 10–12]

These previous findings demonstrate that adolescent pregnancy results from a complex interaction of economic, sociocultural, and personal factors. Understanding these different determinants can help identify the groups of adolescents most at risk of becoming pregnant. It can also inform interventions and policies to prevent adolescent pregnancy and improve the sexual and reproductive health of adolescent girls. Furthermore, tracking changes in the determinants over time can provide valuable information for assessing the efficacy of existing interventions and programmes to reduce adolescent pregnany.

In Ghana, not many studies have explored the predictors of adolescent pregnancy, despite the need for country-specific estimates and determinants to inform national policies and programmes on adolescent sexual and reproductive health. Moreover, the few studies conducted over the last few years had small samples and reported contradictory findings.[13 14] The present study used data from 11 nationally representative surveys conducted between 1988 and 2019 to estimate the national and regional prevalence of adolescent pregnancy and track changes in the sociodemographic and sexual risk behaviours associated with adolescent pregnancy.

## METHODS
Eleven nationally representative cross-sectional surveys from the Ghana Demographic and Health Survey (1988, 1993, 1998, 2003, 2008, 2014), Multiple Indicator Cluster Survey (2006, 2011, 2017–2018) and Malaria Indicator Survey (2016 and 2019) provided the data for this analysis. A detailed description of the methods used in the surveys, including information about survey design, sampling and data collection, is published elsewhere.[15] Briefly, the surveys used a multistage stratified cluster sampling method to select households from rural and urban clusters in Ghana's former 10 administrative regions (now 16 regions). The participant questionnaires administered

in these surveys are comparable, allowing data pooling across surveys and time. For all the surveys, trained field officers under the supervision of the Ghana Statistical Services administered the questionnaires face-to-face to women and men in selected households. The response rate in the surveys was higher than 90%. Data on only women aged 15–19 years were extracted from data files of the 11 surveys for the present analysis.

## Variables
The outcome of interest was adolescent pregnancy. In this study, it was defined as women aged 15–19 who had previously given birth or were pregnant at the time of the survey. Across the surveys, the same questions were asked to elicit responses to measure the outcome. Women were asked, 'are you pregnant now?' to determine those pregnant at the time of the survey and 'have you ever given birth?' to ascertain those who had ever given birth. Response options for both questions were 'yes' or 'no'. In order to reduce reverse causation and recall bias, the analysis of the factors associated with adolescent pregnancy was restricted to adolescents who were pregnant at the time of the survey. Adolescent sociodemographic and sexual risk behaviours were explored as potential explanatory factors of adolescent pregnancy. The sociodemographic factors included adolescent age, place of residence, sex of household head, number of household members ('≤4 members' and '>4 members'[16]), mother still alive, father still alive, literacy level (illiterate='cannot read at all', semi-literate='able to read only partially sentence' and literate='able to read whole sentence'), employment status, household income and frequency of media exposure (newspaper+radio+TV). The sexual behaviours that were considered in the analysis are adolescents' age at first sex ('≤14 years' vs '≥ 15 years'[17]), number of previous sex partners, age of last sexual partner, knowledge of contraception and knowledge of fertility period during menstruation.

## Data management and statistical analysis
Data from each survey year were standardised and pooled into a single dataset before being grouped into three periods, 1988–1998, 2003–2008 and 2011–2019. The prevalence of adolescent pregnancy was estimated as a percentage of all adolescents aged 15–19 years for each survey year, the three eras and the overall sample. Time trends of adolescent pregnancy and age patterns were plotted for survey years and selected sociodemographic factors. A random effect meta-analysis was used to pool the prevalence across the survey years to estimate the prevalence of adolescent pregnancy in Ghana. The pooled prevalence and corresponding 95% CIs were presented using forest plots. Heterogeneity among the surveys was assessed using Higgins and Thompson's $I^2$ statistic.[18] Subgroup analyses were performed to assess variations in the prevalence across the administrative regions of Ghana and rural–urban residence. To scale prevalence

estimates to represent all adolescents in Ghana, denormalised survey weights were applied to correct for unequal sampling in certain areas and subgroups.

Bivariate logistic regression was used to investigate the unadjusted association between the explanatory variables and the binary outcome, adolescent pregnancy. Multivariable logistic regression was then used to determine the factors associated with adolescent pregnancy across the three decades (1988–1998, 2003–2008 and 2011–2019) and the overall sample (1988–2019) in an adjusted analysis. Potential determinants were categorised into distal (sociodemographic factors) and proximal (sexual behaviours) factors, and a conceptual framework was used to guide the analysis (online supplemental figure 1). Based on the framework, distal factors were considered potential confounders of the association between the proximal factors and the outcome in the multivariable model. The logistic regression models accounted for stratification in the survey design, the year surveys were conducted and calculated robust standard errors to account for clustering in the design of the surveys. To minimise the influence of reverse causation and recall bias, the logistic regression analysis was restricted to adolescents who were pregnant at the time of the survey. Because this study is exploratory, $p<0.10$ was used to determine statistically significant associations. A test for linear trend was performed for 'literacy level', 'household income' and 'number of previous sex partners'. Estimates for explanatory variables not available in all the surveys were produced in a separate multivariable model. Stata V.17 was used for all data management and analysis.

### Patient and public involvement
Patients or the public were not involved in the design, conduct, reporting or dissemination plans of our research.

## RESULTS
### Characteristics of the study sample
A nationally representative sample of 14 556 adolescents aged 15–19 years from surveys conducted between 1988 and 2019 was included in the analysis. Table 1 presents the characteristics of the total study sample. Most adolescents lived in rural areas, in households with high income, headed by men and with more than five household members. The majority of the sample were middle adolescents aged 15–17 years. Only a small percentage of the adolescents were orphans or worked at some point in the 12 months before the surveys. Nearly a third (28%) of the adolescents were illiterate, and 63.1% were exposed to mainstream media at least once a week (radio, television or newspaper). Three-fourths (75.9%) of the adolescent-initiated sex after age 14, and 66.7% had had only one sexual relationship. Most adolescents were aware of modern or traditional contraceptive methods, and for the majority of those who had had a sexual partner, their last sexual partner was an adult over age 20.

**Table 1** Characteristics of the study population, 1988–2019 (n=14 556)

| | Unweighted number | Weighted per cent |
|---|---|---|
| **Age** | | |
| 15–17 | 8986 | 62.4 |
| 18–19 | 5570 | 37.6 |
| **Place of residence** | | |
| Urban | 6254 | 47.1 |
| Rural | 8302 | 52.9 |
| **Sex of household head** | | |
| Male | 8718 | 59.9 |
| Female | 4989 | 40.1 |
| **Number of household members** | | |
| ≤4 members | 4209 | 30.8 |
| >4 members | 10 347 | 69.2 |
| **Mother still alive** | | |
| Yes | 5454 | 94.2 |
| No | 309 | 5.8 |
| **Father still alive** | | |
| Yes | 4965 | 86.1 |
| No | 790 | 13.9 |
| **Literacy level** | | |
| Illiterate | 3540 | 27.9 |
| Semi-literate | 1932 | 14.1 |
| Literate | 6238 | 58.1 |
| **Employment** | | |
| Yes | 1985 | 30.3 |
| No | 4442 | 69.7 |
| **Household income** | | |
| Low income | 6268 | 36.9 |
| Middle income | 2561 | 21.2 |
| High income | 4878 | 41.9 |
| **Frequency of media exposure** | | |
| Not at all | 2483 | 22.1 |
| Less than once a week | 1282 | 14.8 |
| At least once a week | 5134 | 63.1 |
| **Age at first sex** | | |
| ≤14 years | 1107 | 24.1 |
| ≥15 years | 3648 | 75.9 |
| **Number of previous sex partners** | | |
| 1 | 764 | 66.7 |
| 2 | 243 | 23.8 |
| Three or more | 106 | 9.4 |
| **Age of last sexual partner** | | |
| Below 20 years | 381 | 28.4 |
| More than 20 years | 993 | 71.6 |

Continued

**Table 1** Continued

| | Unweighted number | Weighted per cent |
|---|---|---|
| **Knowledge of contraception** | | |
| Knows no method | 666 | 14.1 |
| Knows modern or traditional method | 4892 | 85.9 |
| **knowledge of fertility period during menstruation** | | |
| Just before or during menstrual period | 546 | 14.6 |
| Right after menstrual period or at any time | 1964 | 52.1 |
| Halfway between two menstrual periods | 1067 | 33.3 |

### Prevalence and time trends of adolescent pregnancy, 1988–2019

Based on a meta-analysis of the 11 national surveys, the pooled prevalence of adolescent pregnancy was 15.4% (95% CI=13.49% to 17.30%) with significant variation across the surveys ($I^2$=89.9%, p<0.001; figure 1). There was a substantial regional disparity in the prevalence, ranging from 8.2% (95% CI=6.29% to 10.39%) in the Greater Accra region to 19.3% (95% CI=15.75% to 23.19%) in the Central region (table 2; see online supplemental figure 2 for details). Table 2 shows that the pooled prevalence of adolescent pregnancy in rural areas (19.5%) was nearly twice that of urban areas (10.6%) (table 2; see online supplemental figure 3 for details).

The prevalence of adolescent pregnancy declined between 1988 and 1998, then remained relatively consistent from 2003 to 2008 before gradually increasing in the last decade (figure 2A). Adolescents from high-income households (figure 2B) and those who were literate (figure 2C) had a lower prevalence of adolescent pregnancy throughout the 11 surveys, with the steepest decline seen between 1993 and 2006. The surveys showed an upward trend of adolescent pregnancy with increasing age (figure 2D). There was no consistent pattern in the prevalence of adolescent pregnancy across ethnic groups.

### Sexual behaviours and sociodemographic factors associated with adolescent pregnancy

Online supplemental table 1 summarises the sexual behaviour and sociodemographic characteristics according to the number and percentage of adolescents

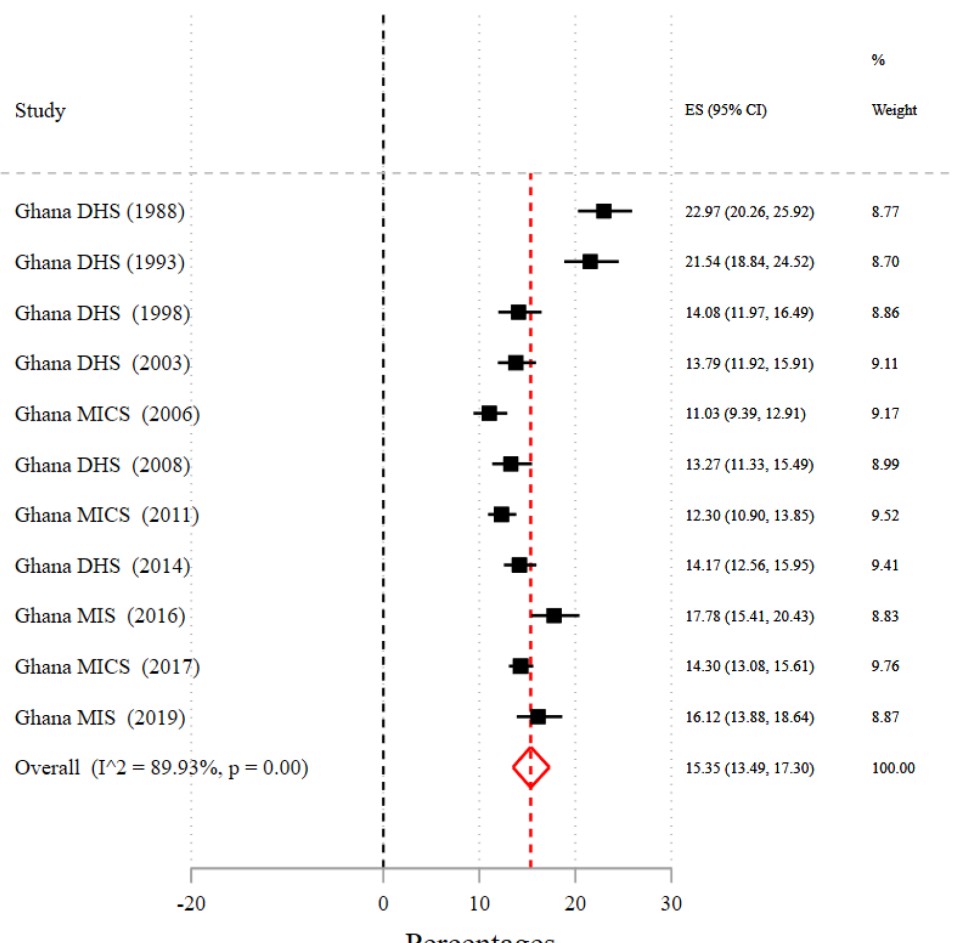

**Figure 1** National prevalence of adolescent pregnancy, 1988–2019. DHS, Demographic and Health Survey; MICS, Multiple Indicator Cluster Survey; MIS, Malaria Indicator Survey.

**Table 2** Subnational prevalence of adolescent pregnancy based on a meta-analysis of nationally representative surveys from 1988 to 2019

|  | Prevalence of adolescent pregnancy | 95% confidence interval | I² |
|---|---|---|---|
| **Region** | | | |
| Western region | 14.42% | 11.27% to 17.87% | 66.5% |
| Central region | 19.34% | 15.75% to 23.19% | 65.0% |
| Greater Accra region | 8.23% | 6.29% to 10.39% | 62.6% |
| Volta region | 16.18% | 13.50% to 19.05% | 36.8% |
| Eastern region | 16.20% | 13.93% to 18.61% | 31.6% |
| Ashanti region | 14.14% | 11.94% to 16.48% | 60.7% |
| Brong-Ahafo region | 16.73% | 12.66% to 21.22% | 75.8% |
| Northern region | 16.41% | 11.62% to 21.82% | 79.7% |
| Upper West region | 11.48% | 8.48% to 14.82% | 0.0% |
| Upper East region | 11.40% | 8.73% to 14.36% | 0.0% |
| **Residence** | | | |
| Urban | 10.59% | 8.69% to 12.66% | 85.6% |
| Rural | 19.50% | 17.51% to 21.57% | 79.3% |

See online supplemental figures 2 and 3 for detailed forest plots of each region and urban–rural residence, respectively.
I² measures the percentage of total variability that was due to between survey heterogeneity.

pregnant at the time of the survey. Results of unadjusted analysis of the association between sociodemographic factors, sexual behaviours and adolescent pregnancy are presented in online supplemental tables 2 and 3, respectively.

In the overall sample, adolescent age, place of residence, household size, literacy level, household wealth, number of previous sex partners and knowledge of contraception were the factors associated with adolescent pregnancy after adjusting for confounders (tables 3 and 4). Middle adolescents (age 15–17 years) were 70% less likely to become pregnant than late adolescents (aOR=0.30, 95% CI=0.23 to 0.39). Adolescents in urban areas were less likely to become pregnant than those in rural areas (aOR=0.56, 95% CI=0.43 to 0.74). The odds of adolescent pregnancy was lower among adolescents in large households (>4 household members) than among those in small households (aOR=0.62, 95% CI=0.49 to 0.78). There was a dose–response relationship between literacy level and adolescent pregnancy. Semiliterates (aOR=0.56, 95% CI=0.39 to 0.82) and literates (aOR=0.28, 95% CI=0.21 to 0.37) were less likely to become pregnant than illiterate. Adolescents who were not working had a reduced likelihood (aOR=0.62, 95% CI=0.43 to 0.90) of becoming pregnant than those who worked. There was a dose–response relationship between household income and adolescent pregnancy. Those from middle-income (aOR=0.91, 95% CI=0.67 to 1.24) and high-income (aOR=0.59, 95% CI=0.36 to 0.94) households were less likely to become pregnant compared with those from low-income households. Similarly, a dose–response relationship was found between the number of previous sex partners and adolescent pregnancy. Adolescents with one

(aOR=0.46, 95% CI=0.23 to 0.95) or two (aOR=0.67, 95% CI=0.33 to 1.36) previous sex partners were less likely to become pregnant than those with three or more previous partners. Adolescents who were unaware of contraceptive methods were less likely (aOR=0.49, 95%CI=0.27 to 0.90) to become pregnant than those who knew about modern or traditional methods.

However, not all the determinants identified in the overall sample were consistently associated with adolescent pregnancy in the last three decades. Between 1988 and 1998, adolescent age, literacy level, employment status, household size and whether the mother was alive were the only factors associated with adolescent pregnancy in the adjusted analysis (tables 3 and 4). Middle adolescents were less likely to become pregnant than late adolescents. Semiliterate or literate adolescents were less likely to become pregnant than illiterate. Those who had not worked in the previous 12 months were less likely than those who had worked to become pregnant. Adolescents in large households were less likely to become pregnant than those in small households. Adolescents whose mothers were still alive were less likely to experience pregnancy than those whose mothers had passed away.

Between 2003 and 2008, adolescent age, literacy level, household size, household wealth, age of last sexual partner, number of previous sex partners and knowledge of contraception were the factors associated with adolescent pregnancy after adjusting for potential confounders (tables 3 and 4). Adolescents in middle-income and high-income families were less likely than those in low-income families to become pregnant. Middle adolescents were less likely to become pregnant than late adolescents. Adolescent pregnancy was less likely among those in large

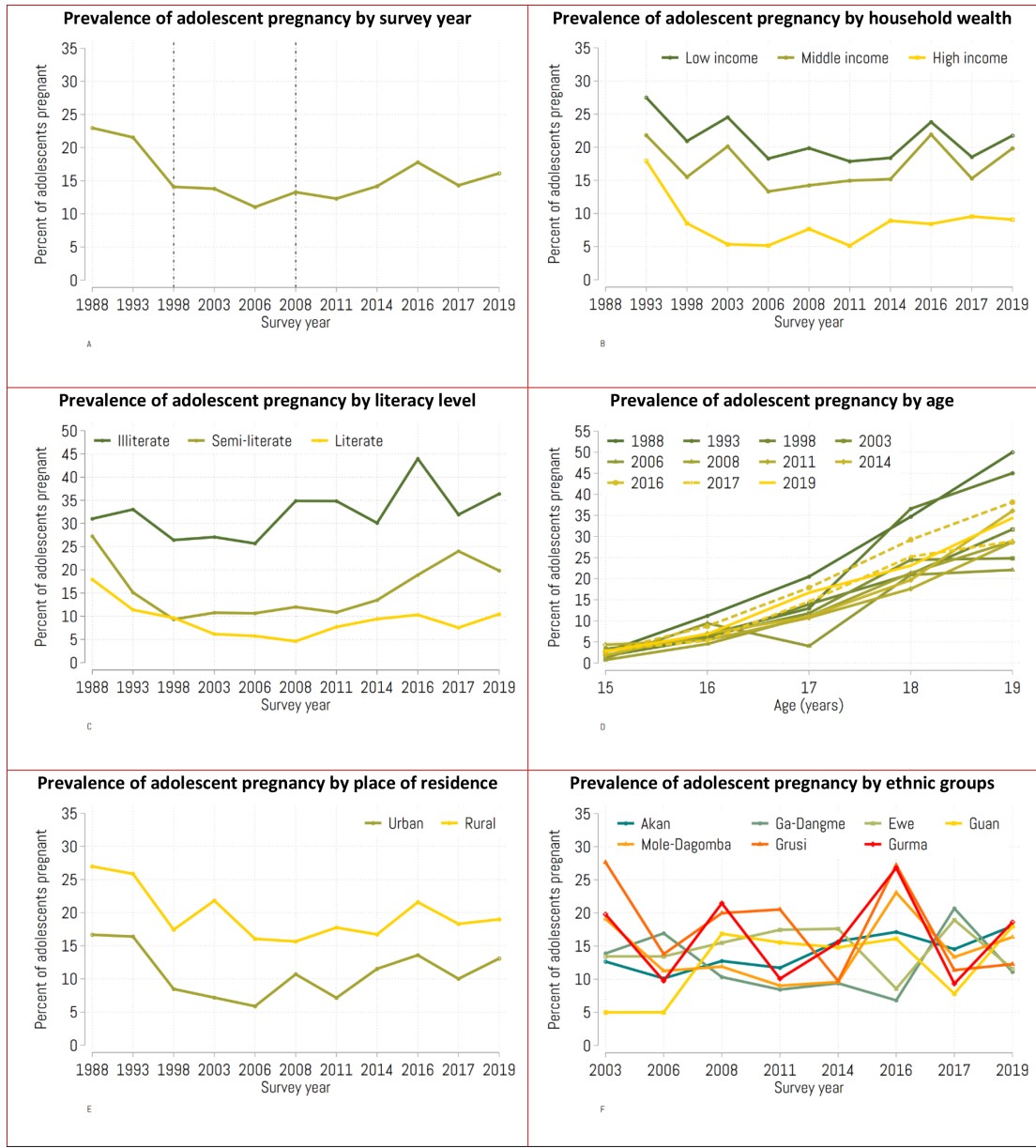

**Figure 2** Time trends of adolescent pregnancy by (A) survey year, (B) household income, (C) literacy level, (D) adolescent age, (E) residence and (F) ethnic group.

households than those in small households. Semiliterates or literates were less likely to become pregnant than illiterate. Those whose last sexual partner was an older person were less likely to become pregnant than those whose last partner was an adolescent. Adolescent pregnancy was less likely among those who had one or two previous sex partners compared with those who had three or more previous partners. Those unaware of contraceptive methods had reduced odds of getting pregnant than those who knew about modern or traditional methods.

From 2011 to 2019, adolescent age, place of residence, literacy level and knowledge of the fertility period during menstruation were the factors associated with adolescent pregnancy after adjusting for other sexual behaviours and sociodemographic factors (tables 3 and 4). Middle adolescents were less likely to become pregnant than late

adolescents. Adolescents in urban areas were about 50% less likely than those in rural areas to become pregnant. Semiliterates and literates were less likely to become pregnant than illiterate. Adolescents who believed the most fertile period was just before or during menstruation or right after it were less likely to become pregnant than those who said it was halfway between two menstrual periods.

## DISCUSSION

Based on a meta-analysis of 11 nationally representative surveys conducted between 1988 and 2019, the pooled prevalence of adolescent pregnancy was 15.4%, with substantial regional variation. The prevalence was higher in rural areas than in urban areas. Over the last three

**Table 3** Results of adjusted logistic regression analysis of the association between sociodemographic characteristics and adolescent pregnancy, 1988–2019

| Survey years | 1988–1998 | 2003–2008 | 2011–2019 | Overall sample (1988–2019) |
|---|---|---|---|---|
| Population analysed (unweighted) | 2541 | 3350 | 8665 | 14 556 |
| Number currently pregnant (unweighted) | 96 | 117 | 282 | 495 |
| | Adjusted odds ratio (95% CI) | Adjusted odds ratio (95% CI) | Adjusted odds ratio (95% CI) | Adjusted odds ratio (95% CI) |
| **Age** | P<0.001 | P<0.001 | P<0.001 | P<0.001 |
| 15–17 | 0.40 (0.24 to 0.66) | 0.39 (0.24 to 0.64) | 0.22 (0.15 to 0.33) | 0.30 (0.23 to 0.39) |
| 18–19 | 1.00 | 1.00 | 1.00 | 1.00 |
| **Place of residence** | P=0.23 | P=0.72 | P=0.002 | P<0.001 |
| Urban | 0.74 (0.45 to 1.22) | 0.86 (0.40 to 1.89) | 0.54 (0.36 to 0.81) | 0.56 (0.43 to 0.74) |
| Rural | 1.00 | 1.00 | 1.00 | 1.00 |
| **Sex of household head** | P=0.23 | P=0.06 | P=0.60 | P=0.42 |
| Male | 1.61 (0.73 to 3.54) | 0.61 (0.36 to 1.02) | 0.91 (0.63 to 1.31) | 0.89 (0.66 to 1.19) |
| Female | 1.00 | 1.00 | 1.00 | 1.00 |
| **Number of household members** | P=0.001 | P=0.02 | P=0.19 | P<0.001 |
| ≤4 members | 1.00 | 1.00 | 1.00 | 1.00 |
| >4 members | 0.46 (0.29 to 0.73) | 0.58 (0.37 to 0.91) | 0.78 (0.54 to 1.13) | 0.62 (0.49 to 0.78) |
| **Mother still alive** | P=0.07 | | P=0.48 | P=0.94 |
| Yes | 0.30 (0.08 to 1.09) | – | 0.58 (0.13 to 2.58) | 0.94 (0.24 to 3.77) |
| No | 1.00 | – | 1.00 | 1.00 |
| **Father still alive** | P=0.85 | P=0.86 | P=0.99 | P=0.84 |
| Yes | 1.00 | 1.00 | 1.00 | 1.00 |
| No | 1.12 (0.36 to 3.44) | 1.13 (0.30 to 4.21) | 1.00 (0.40 to 2.51) | 1.08 (0.52 to 2.24) |
| **Literacy level** | P=0.001* | P<0.001* | P<0.001* | P<0.001* |
| Illiterate | 1.00 | 1.00 | 1.00 | 1.00 |
| Semi-literate | 0.55 (0.20 to 1.50) | 0.42 (0.22 to 0.81) | 0.66 (0.39 to 1.13) | 0.56 (0.39 to 0.82) |
| Literate | 0.40 (0.23 to 0.69) | 0.27 (0.16 to 0.45) | 0.27 (0.17 to 0.42) | 0.28 (0.21 to 0.37) |
| **Employment** | P=0.05 | P=0.29 | P=0.12 | P=0.01 |
| Yes | 1.00 | 1.00 | 1.00 | 1.00 |
| No | 0.61 (0.37 to 1.01) | 0.73 (0.41 to 1.31) | 0.59 (0.30 to 1.15) | 0.62 (0.43 to 0.90) |
| **Household income** | P=0.56 | P<0.001* | P=0.27 | P=0.03* |
| Low income | 1.00 | 1.00 | 1.00 | 1.00 |
| Middle income | 0.67 (0.32 to 1.39) | 0.65 (0.39 to 1.09) | 1.09 (0.70 to 1.69) | 0.91 (0.67 to 1.24) |
| High income | 0.88 (0.45 to 1.74) | 0.30 (0.15 to 0.58) | 0.63 (0.32 to 1.26) | 0.59 (0.36 to 0.94) |
| **Frequency of media exposure** | | P=0.82 | P=0.23 | P=0.51 |
| Not at all | – | 1.00 | 1.00 | 1.00 |
| Less than once a week | – | 1.40 (0.45 to 4.37) | 1.29 (0.62 to 2.69) | 1.25 (0.68 to 2.32) |
| At least once a week | – | 1.04 (0.49 to 2.20) | 1.70 (0.91 to 3.18) | 1.34 (0.82 to 2.20) |

*Test for linear trend

decades, adolescent pregnancy peaked in 1988, followed by a progressive decline in the 1990s and early 2000s; but the prevalence has slowly increased over the last decade.

The pooled data suggest that the prevalence of adolescent pregnancy rose with increasing adolescent age, a pattern consistent across all 11 surveys. The prevalence

Table 4    Results of adjusted logistic regression analysis of the association between sexual behaviours and adolescent pregnancy, 1988–2019.

| Survey years | 1988–1998 | 2003–2008 | 2011–2019 | Overall sample (1988–2019) |
|---|---|---|---|---|
| Population analysed (unweighted) | 2541 | 3350 | 8665 | 14556 |
| Number currently pregnant (unweighted) | 96 | 117 | 282 | 495 |
| | Adjusted odds ratio (95% CI) | Adjusted odds ratio (95% CI) | Adjusted odds ratio (95% CI) | Adjusted odds ratio (95% CI) |
| **Age at first sex** | P=0.63 | P=0.79 | P=0.10 | P=0.14 |
| ≤14 years | 0.85 (0.45 to 1.63) | 0.91 (0.44 to 1.88) | 0.63 (0.36 to 1.10) | 0.75 (0.52 to 1.10) |
| ≥15 years | 1.00 | 1.00 | 1.00 | 1.00 |
| **Number of previous sex partners** | | P=0.03* | P=0.79 | P=0.04* |
| 1 | – | 0.29 (0.10 to 0.88) | 0.73 (0.27 to 1.95) | 0.46 (0.23 to 0.95) |
| 2 | – | 0.56 (0.21 to 1.52) | 0.90 (0.31 to 2.58) | 0.67 (0.33 to 1.36) |
| Three or more | – | 1.00 | 1.00 | 1.00 |
| **Age of last sexual partner** | | P=0.02 | P=0.20 | P=0.39 |
| Below 20 years | – | 1.00 | 1.00 | 1.00 |
| More than 20 years | – | 0.14 (0.03 to 0.76) | 2.00 (0.69 to 5.81) | 1.46 (0.61 to 3.48) |
| **Knowledge of contraception** | P=0.20 | P=0.08 | | P=0.02 |
| Knows no method | 0.62 (0.30 to 1.29) | 0.36 (0.11 to 1.15) | – | 0.49 (0.27 to 0.90) |
| Knows modern or traditional method | 1.00 | 1.00 | – | 1.00 |
| **knowledge of fertility period during menstruation** | P=0.56 | P=0.45 | P=0.02 | P=0.83 |
| Just before or during menstrual period | 2.09 (0.50 to 8.67) | 1.88 (0.70 to 5.05) | 0.17 (0.04 to 0.69) | 0.90 (0.45 to 1.77) |
| Right after menstrual period or at any time | 0.95 (0.42 to 2.13) | 1.24 (0.60 to 2.56) | 0.51 (0.24 to 1.07) | 0.87 (0.56 to 1.36) |
| Halfway between two menstrual periods | 1.00 | 1.00 | 1.00 | 1.00 |

*Test for linear trend

reported in this study is lower than the 18.8%, 19.3%, and 17.7% prevalence estimates for Africa, sub-Saharan Africa and West Africa, respectively.[3]

In line with the present findings, studies in Indonesia and Uganda reported a higher likelihood of pregnancy among teenagers living in rural areas, those with no or low education, and adolescents in low-income households.[19 20] Similarly, in a systematic review of 24 studies in sub-Saharan Africa, about half of the included studies reported a higher likelihood of adolescent pregnancy in low-income households.[11] Furthermore, an analysis of the 2014–2015 Rwanda Demographic and Health Survey confirmed that adolescents from large families were less likely than those from small families to get pregnant.[21] Similar to the findings of this study, Jonas et al, found higher odds of adolescent pregnancy among those with more than one sex partner in South Africa.[22] Though there was no association between adolescent age and pregnancy in the South African study, in a pooled analysis of data from five African countries and two cross-sectional studies in Ghana, older adolescents had higher odds of adolescent pregnancy.[7 13 23] In support of the present findings, a qualitative study in Kenya found that adolescents engaged in incoming generating activities were more likely to get pregnant than those not working[24]; however, a study in Ethiopia reported a contrary finding.[25] In an earlier study in Ghana, there was no association between care status (orphanhood) and adolescent pregnancy,[23] which agrees with the current results for 2011–2019 and the overall sample.

There are several reasons to explain the heightened vulnerability of adolescents in low-income households to pregnancy. In most cases, adolescents in these homes are not in school; instead, parents force them into early marriage to relieve the financial burden of taking care of the adolescents or prevent out-of-wedlock pregnancy.[26] Several studies have revealed that most adolescent girls from poor homes engage in early sexual relationships for financial and material gains because parents cannot provide their basic needs, including food, toiletries, clothes, shoes, school supplies, fees and housing.[24 27–30] For instance, in a Ugandan study, adolescents acknowledged that they sometimes offer sex to boys/men as payment for money borrowed or in return for material items because their parents cannot provide.[30] Under these situations, adolescents are less able to negotiate safe sex and condom use, increasing their chance of pregnancy. Some may engage multiple sexual partners for greater financial benefits. Therefore, it is not surprising that in this study, adolescents with three or more sexual partners were more likely to experience pregnancy than those with one partner.

Additionally, some adolescents engage in all manner of income-generating activities to meet their basic needs, and this exposes them to mistreatment, sexual abuse and exploitation, particularly those working as domestic help, bartenders, waitresses, shopkeepers and selling at night or work that involves late-night shifts.[24 30] This is probably why adolescents who worked prior to the surveys analysed in this study had a higher chance of becoming pregnant. Also, it has been suggested that some adolescents spend the income from these jobs on activities that increase their risk of pregnancy, such as alcohol, drugs and nightclubs.[24] In large families, adolescents are supervised, guided and supported emotionally and financially not only by their parents but also by other family members to make sexual and reproductive health decisions that favour pregnancy prevention, reducing the likelihood of adolescent pregnancy in such families as reported in this study. Also, in these families' adolescents who are uncomfortable discussing sexual health issues with their parents have other family members to consult.

The lower likelihood of pregnancy among adolescents in urban areas may be due to the advantages the urban setting offers. Urban residents have easy access to pregnancy prevention information and services, including easy access to various effective modern contraceptive methods (including emergency contraceptives) through the many hospitals, clinics, pharmacy shops and family planning centres in urban areas. In rural areas, misconceptions are promoted, even when contraceptives are available, and sexually active adolescents usually rely on traditional contraceptive methods that are ineffective. Furthermore, traditional and cultural practices that promote early adolescent marriage are rarely practised in most urban settings, reducing adolescents' exposure to pregnancy. In addition, most adolescents in urban areas are better educated than their rural counterparts, increasing their ability to understand and effectively use sexual and reproductive health information to reduce their risk of pregnancy. This further explains why adolescents who were semiliterate or literate in this study were less likely to experience pregnancy than those who were illiterate. Also, schooling delays marriage and empowers girls to make rational decisions in their relationships that prevent pregnancy.[10]

Most adolescents in the present study did not know the fertile phase of the menstrual cycle. Therefore, it is surprising that those who did not know the fertile period were less likely to become pregnant. It is possible that these girls had not initiated sex or that those who correctly identified the fertile phase learnt it when they became pregnant (reverse causality). This may also explain the unexpected finding on knowledge about contraceptives. However, it was not surprising that older adolescents were more likely to get pregnant than younger adolescents because most older adolescents are sexually active and engage in more sexual risk behaviours than middle adolescents. It is also possible that parents and other adults support and guide middle-aged adolescents more

than they do older teenagers. In addition, the lower adolescent pregnancy among those whose last partner was older in this study is possible due to better pregnancy prevention and sexual and reproductive health knowledge of older partners.

Age-appropriate sexual and reproductive health education as part of the school curriculum and in communities is essential to help adolescents from all settings, income groups and family backgrounds make rational decisions that will delay pregnancy and motherhood. Adolescents access and use of contraceptives, including emergency contraceptives, should be promoted, particularly in rural communities and low-income households. Traditional leaders and stakeholders who promote cultural and traditional practices that encourage early motherhood should be educated on the harmful effects of adolescent childbearing, and such practices discouraged. Girls should be encouraged and motivated to stay in school even when they become pregnant, considering the long-term repercussions of dropping out. In addition, interventions to reduce adolescent pregnancy should include boys and men.

The main strength of this study is the use of a large, nationally representative sample that captures adolescents from various sociodemographic backgrounds and settings. The study's findings are generalisable to all adolescents in Ghana and more relevant to national policies than previous studies that mainly focused on individual communities. One major limitation of the study is the possibility of reverse causality and recall bias. To minimise the effect of reverse causality, analysis of the determinants was restricted to currently pregnant adolescents. However, it is important to note that this study was not designed for causal relationships, so I make no such claims. Furthermore, it is possible that the prevalence reported in this study was underestimated because most Ghanaian women admit to pregnancy after the first trimester. In addition, the definition of adolescent pregnancy in this study excluded abortion. Furthermore, because the study relied on secondary data, cultural and traditional practices that could potentially impact adolescent pregnancy were not considered in the analysis because data were unavailable on these practices. Future studies should address the limitations with longitudinal and qualitative studies.

## CONCLUSION

Adolescent pregnancy has risen slowly over the last decade, following a steady prevalence between 1998 and 2008. Regional and urban–rural disparities in the prevalence were substantial, and age patterns showed that adolescent pregnancy increased as adolescents got older. The factors influencing adolescent pregnancy differed across the eras except for literacy level and adolescent age; illiteracy and older adolescent age were associated with a higher likelihood of adolescent pregnancy across the three decades from 1988 to 2019.

**Acknowledgements** The author would like to thank measure DHS and UNICEF's MICS programme for prompt approval and access to the datasets for the analysis.

**Contributors** SM conceived and designed the study, accessed the data, conducted the literature search, performed the statistical analysis and drafted the manuscript. SM is responsible for the overall content.

**Funding** This study was supported by the Economic and Social Research Council (ESRC) (grant number: ES/P000592/1). The ESRC had no role in the study's design, analysis of the data or manuscript preparation.

**Competing interests** None declared.

**Patient and public involvement** Patients and/or the public were not involved in the design, or conduct, or reporting, or dissemination plans of this research.

**Patient consent for publication** Not applicable.

**Ethics approval** The Ghana Statistical Services and DHS programme obtained ethical approval, participant consent, and permission for each primary survey. MEASURE DHS granted permission and access for the secondary use of the survey data for the present analysis. Because the data is available in the public domain with no personal identifiers, ethical approval was not sought for the secondary analysis.

**Provenance and peer review** Not commissioned; externally peer reviewed.

**Data availability statement** Data may be obtained from a third party and are not publicly available. All the datasets analysed are publicly available with the measure DHS program (https://dhsprogram.com) and UNICEF's MICS programme (https://mics.unicef.org/).

**ORCID iD**
Shamsudeen Mohammed http://orcid.org/0000-0002-3771-8425

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
