## [Reviewer comments · BMJ Open]

ARTICLE DETAILS

TITLE (PROVISIONAL)	National and subnational prevalence of adolescent pregnancy and changes in the associated sexual behaviours and sociodemographic determinants across three decades in Ghana, 1988 – 2019: analysis of national surveys
AUTHORS	Mohammed, Shamsudeen

VERSION 1 – REVIEW

REVIEWER	Mezmur, Haymanot Haramaya University College of Health and Medical Sciences, Public Health
REVIEW RETURNED	13-Dec-2022

GENERAL COMMENTS	The study is not Met-analysis but the comparison was made with those meta analysis studies and also the source of data is not clearly mentioned.
--

REVIEWER	Ewemooje, Olusegun University of Botswana, Department of Statistics
REVIEW RETURNED	17-Dec-2022

GENERAL COMMENTS	Comments The paper makes a good read and innovative as it seeks to examine changes over time in the prevalence of adolescent pregnancy and its associated sexual behaviours and sociodemographic determinants in Ghana. However, the following are my concerns: Major comments The outcome variable is not adequately defined under methods. For instance, how was the response variable for male respondents generated? As male can not be asked if they are pregnant or not? Also, male can not be asked if they have given birth? On the explanatory variable, "Age at first sex" was categorized as – before age 16 and 16-19 years. This may not be appropriate as the conventional way of categorizing this is; early sexual debut (i.e. have sex on or before age 14years) or otherwise. Check Biney, et al., (2022): Predictors of sexual risk behaviour among unmarried persons aged 15-34 years in South Africa, The Social Science Journal, 59:4, 543-558, https://doi.org/10.1080/03623319.2020.1727225 and Amoateng, et al., (2022): Prevalence and determinants of adolescent pregnancy among women of reproductive age in South Africa. African Journal of Reproductive Health January, 26 (1):82-91. https://doi.org/10.29063/ajrh2022/v26i1.9 for possible help on this. The use of literacy and level of education as different variables is inappropriate as both measures the same or almost the same thing. Thus, this may reduce the effect of other variables, hence, one of
---

	the variables should be removed. The last sentence under “variables” reads - The explanatory variables were not available for all surveys. So, how did you tackle this problem? Under Results How did you move from the descriptive analysis to predictive analysis? I will suggest you test the association between your response variable and individual explanatory variable first. Afterwhich, significant explanatory variables will then be used in the logistic regression models. Minor Comments No consistency in the population analysed (n) in Table 3 and Table 4. I think you may have to look at Table 3 carefully. Results on the overall (1988-2019) was omitted under “Sexual behaviours and sociodemographic factors associated with adolescent pregnancy” The manuscript needs to be properly proofread Decision The manuscript should be allowed to go through major revision.
--	--

VERSION 1 – AUTHOR RESPONSE

Reviewer: 1

Mrs. Haymanot Mezmur, Haramaya University College of Health and Medical Sciences

Comments to the Author:

The study is not Met-analysis but the comparison was made with those meta analysis studies and also the source of data is not clearly mentioned.

Response: Your comments are very much appreciated. I agree with the reviewer that the study is not a meta-analysis but a secondary analysis that employed meta-analysis techniques. So no claims are made in the study to suggest that it was a meta-analysis of primary studies. Regarding the source of the data not clearly mentioned, an attempt has been made in the abstract and methods section to address this as below:

In the abstract:

Eleven nationally representative cross-sectional surveys from the Ghana Demographic and Health Survey (1988, 1993, 1998, 2003, 2008, 2014), Multiple Indicator Cluster Survey (2006, 2011, 20117-18), and Malaria Indicator Survey (2016 and 2019) provided data on 14556 adolescents aged 15-19 for this analysis.

In the methods section:

Eleven nationally representative cross-sectional surveys from the Ghana Demographic and Health Survey (1988, 1993, 1998, 2003, 2008, 2014), Multiple Indicator Cluster Survey (2006, 2011, 20117-18), and Malaria Indicator Survey (2016 and 2019) provided the data for this analysis.

Reviewer: 2

Dr. Olusegun Ewemooje, University of Botswana, Federal University of Technology Akure

Comments to the Author:

Comments

The paper makes a good read and innovative as it seeks to examine changes over time in the prevalence of adolescent pregnancy and its associated sexual behaviours and sociodemographic

determinants in Ghana. However, the following are my concerns:

Major comments

The outcome variable is not adequately defined under methods. For instance, how was the response variable for male respondents generated? As male can not be asked if they are pregnant or not? Also, male can not be asked if they have given birth?

Response: Thank you for the insightful comment. In the methods section, the outcome was defined as "...women aged 15 to 19 years who had a birth or were pregnant at the time of the interview". This means that only females were included in the analysis. The DHS does not ask males about pregnancy experiences. To make this clearer, I have now rewritten the sentence to read, "*Data on only women aged 15 to 19 years were extracted from data files of the eleven surveys for the present analysis*" to clarify that the study sample included only females.

On the explanatory variable, "Age at first sex" was categorised as – before age 16 and 16-19 years. This may not be appropriate as the conventional way of categorising this is; early sexual debut (i.e. have sex on or before age 14years) or otherwise. Check Biney, et al., (2022): Predictors of sexual risk behaviour among unmarried persons aged 15-34 years in South Africa, *The Social Science Journal*, 59:4, 543-558, <https://doi.org/10.1080/03623319.2020.1727225> and Amoateng, et al., (2022): Prevalence and determinants of adolescent pregnancy among women of reproductive age in South Africa. *African Journal of Reproductive Health* January, 26 (1):82-91. <https://doi.org/10.29063/ajrh2022/v26i1.9> for possible help on this.

Response: Thank you for the suggestions. I have regrouped the variables into ≤ 14 years and ≥ 15 years based on what was done in Amoateng, et al., (2022): Prevalence and determinants of adolescent pregnancy among women of reproductive age in South Africa. *African Journal of Reproductive Health* January, 26 (1):82-91. The paper has been appropriately referenced to support that decision. All the descriptive, unadjusted, and adjusted tables have been updated to reflect the new grouping.

The use of literacy and level of education as different variables is inappropriate as both measures the same or almost the same thing. Thus, this may reduce the effect of other variables, hence, one of the variables should be removed.

Response: Thank you for the suggestion. I have now dropped "educational level" from the analysis and maintained "literacy level" since literacy level is more likely the outcome of the education and a more appropriate measure of an adolescent's ability to use written information.

The last sentence under "variables" reads - The explanatory variables were not available for all surveys. So, how did you tackle this problem?

Response: Thank you for the interesting question. Because the inclusion of such variables had the potential to reduce the power of the study, they were analysed separately. For example, between 1988 and 1998, "father still alive" was available in only 1988. So, it was analysed separately but adjusted for the same confounders as the other variables. In the "**Data management and statistical analysis**" section of the manuscript, I explained that "*Estimates for explanatory variables not available in all the surveys were produced in a separate multivariable model.*" This technique allows the variable to be included in the analysis while maintaining the statistical power to detect a difference. The technique is common in the analysis of data from secondary sources. An example is the analysis done in this paper here <https://www.nature.com/articles/s41586-020-2521-4>.

Under Results

How did you move from the descriptive analysis to predictive analysis? I will suggest you test the association between your response variable and individual explanatory variable first. After which, significant explanatory variables will then be used in the logistic regression models.

Response: Your comment is much appreciated, and apologies for not making this very clear in the methods section. I did exactly what you suggested, and the bivariate analysis results were uploaded as a supplementary document since I have many tables and figures in the main manuscript. I have

now added the sentence "Bivariate logistic regression was used to investigate the unadjusted association between the explanatory variables and adolescent pregnancy" in the "Data management and statistical analysis" section to clarify that a bivariate analysis was done first before the multivariable analysis. Variable selection for multivariable analysis was not based on only P-values. In addition to p-values, I also considered the theoretical importance of each variable and effect size. However, the variables with the smallest p-values were introduced into the multivariable models first.

Minor Comments

No consistency in the population analysed (n) in Table 3 and Table 4. I think you may have to look at Table 3 carefully.

Response: Thank you for the observation. The numbers in Table 3 are just univariate counts that do not consider other variables. However, the numbers in Table 4 show the number of participants left after adjusting for other variables. Because this is a complete case analysis of datasets with some missing values, the numbers in the descriptive tables (Table 4) are not expected to be the same as those in the final adjusted models (Table 4). After adjusting for other variables, some numbers are lost because not all the surveys have complete data on each individual for each variable. This accounts for the differences between the numbers shown in the two tables. I showed the numbers in Table 4 to make clear to the reader the number of participants in the final model, as this is important in deciding whether the final models were sufficiently powered.

Results on the overall (1988-2019) was omitted under "Sexual behaviours and sociodemographic factors associated with adolescent pregnancy"

Response: Thank you for your keen observation and insightful comment. I have now provided the results for the overall sample (1988-2019) under "Sexual behaviours and sociodemographic factors associated with adolescent pregnancy". This interesting comment has also prompted me to revise the results section of the abstract to reflect the findings from the overall sample and different eras. Thank you.

VERSION 2 – REVIEW

REVIEWER	Ewemooje, Olusegun University of Botswana, Department of Statistics
REVIEW RETURNED	27-Feb-2023

GENERAL COMMENTS	Comments The author has improved the paper by addressing almost all the comments. However, the following are still of concerns: Major comments On the explanatory variable, "Age at first sex" was categorized as – before age 16 and 16-19 years. This has been corrected in table 1 but the author forgot to make the same changes in further analysis as shown in table 4. Minor Comments The manuscript needs to be properly proofread. Decision The manuscript should be allowed to go through minor revision.
---

VERSION 2 – AUTHOR RESPONSE

Reviewer: 2

Comments to the Author:

Comments

The author has improved the paper by addressing almost all the comments. However, the following are still of concerns:

Major comments

On the explanatory variable, "Age at first sex" was categorized as – before age 16 and 16-19 years. This has been corrected in table 1 but the author forgot to make the same changes in further analysis as shown in table 4.

Response: Thank you for comment and for drawing my attention to the omitted amendment. I have now made the appropriate changes in Table 4 using the right categories (≤ 14 years vs ≥ 15 years) for "age at first sex".

Minor Comments

The manuscript needs to be properly proofread.

Response: Thank you for your suggestion, which has been accepted. I have proofread the manuscript.